# Same-visit hepatitis C testing and treatment to accelerate cure among people who inject drugs (the QuickStart Study): a cluster randomised cross-over trial protocol

Joseph S Doyle [1,2] Katherine Heath,[2] Imogen Elsum,[2] Caitlin Douglass,[2] Amanda Wade,[2] Jessica Kasza [3] Kate Allardice,[2] Sally Von Bibra,[2] Kico Chan,[2] Beatriz Camesella,[2] Rodney Guzman,[2] Mellissa Bryant,[2] Alexander J Thompson,[4,5] Mark A Stoové,[2] Thomas L Snelling,[6,7] Nick Scott,[2] Timothy Spelman,[2] David Anderson,[2] Jacqui Richmond,[2] Jessica Howell,[2,4] Nada Andric,[8] Paul M Dietze,[2,3] Peter Higgs,[2,9] Rachel Sacks-Davis,[2] Andrew B Forbes,[3] Margaret E Hellard,[2,3] Alisa E Pedrana [2,3]

JSD and KH are joint first authors.

**Correspondence to**
Professor Joseph S Doyle; joseph.doyle@burnet.edu.au

## ABSTRACT

**Introduction** Despite universal access to government-funded direct-acting antivirals (DAAs) in 2016, the rate of hepatitis C treatment uptake in Australia has declined substantially. Most hepatitis C is related to injecting drug use; reducing the hepatitis C burden among people who inject drugs (PWID) is, therefore, paramount to reach hepatitis C elimination targets. Increasing DAA uptake by PWID is important for interrupting transmission and reducing incidence, as well as reducing morbidity and mortality and improving quality of life of PWID and meeting Australia's hepatitis C elimination targets.

**Methods and analysis** A cluster randomised cross-over trial will be conducted with three intervention arms and a control arm. Arm A will receive rapid hepatitis C virus (HCV) antibody testing; arm B will receive rapid HCV antibody and rapid RNA testing; arm C will receive rapid HCV antibody testing and same-day treatment initiation for HCV antibody-positive participants; the control arm will receive standard of care. The primary outcomes will be (a) the proportion of participants with HCV commencing treatment and (b) the proportion of participants with HCV achieving cure. Analyses will be conducted on an intention-to-treat basis with mixed-effects logistic regression models.

**Ethics and dissemination** The study has been approved by the Alfred Ethics Committee (number HREC/64731/Alfred-2020-217547). Each participant will provide written informed consent. Reportable adverse events will be reported to the reviewing ethics committee. The findings will be presented at scientific conferences and published in peer-reviewed journals.

**Trial registration number** NCT05016609.

**Trial progression** The study commenced recruitment on 9 March 2022 and is expected to complete recruitment in December 2024.

## STRENGTHS AND LIMITATIONS OF THIS STUDY

⇒ The QuickStart Trial will recruit from a focused population, people with current or past history of injecting drugs, who are most at risk of hepatitis C.

⇒ A cross-over clustered randomised control trial design will be used to increase study power by leveraging within-site comparisons.

⇒ Four study conditions will be investigated to analyse three rapid point-of-care (POC) testing and treatment interventions, compared with a control arm; one arm investigates a rapid POC antibody test and same-day treatment—an intervention that could be used in resource-limited settings if found to be effective.

⇒ Medical record review will be used in addition to in-person participant follow-up to monitor study outcomes and reduce loss to follow up.

⇒ The participant recruitment strategy counteracts several biases present in observational studies, but participation is voluntary, so a risk of selection bias remains.

## INTRODUCTION

As of 2019, approximately 58 million people worldwide were living with hepatitis C virus (HCV), causing around 290 000 deaths annually.[1] In Australia, approximately 118 000 people were living with HCV at the end of 2020.[2]

Direct-acting antiviral medications (DAAs) are highly effective (cure rates >90%), have minimal side effects and require only 8–12 weeks of daily tablets. The Australia Pharmaceutical Benefits Scheme is a government programme that subsidises medicines to

improve affordability.[3] In 2016, the Pharmaceutical Benefits Scheme negotiated a fixed-price contract providing access to DAAs for every Australian living with HCV, giving Australia a unique opportunity to eliminate HCV.[4 5]

Most untreated HCV infections in Australia are among people with a history of injecting drug use.[2 6] Models of HCV epidemics where transmission is driven by injecting drug use suggest that treating people who inject drugs (PWID) and people with a history of injecting drugs will lead to substantial reductions in HCV incidence and liver-related morbidity and mortality.[7–9] Indeed, models have shown that treating as few as 6% of the estimated population of PWID in Australia annually for 10 years would reduce new HCV infections by 80% and HCV prevalence in PWID by 10%.[10 11] However, in Australia, PWID remain underdiagnosed, undertreated and underserviced in HCV care and healthcare more broadly.[2 12–14]

Unrestricted access to DAAs following government subsidy in 2016 resulted in an initial spike in treatment uptake due to increased availability of the medication to those who had previously been unable to access it. However, treatment numbers have since declined, exacerbated by the COVID-19 pandemic.[2 15 16] There exist service, provider and individual barriers to accessing HCV care.[17] One service-related barrier among PWID is the current need for multiple venepunctures in order to be diagnosed with HCV in Australia, particularly as venous access can be challenging in PWID.[18 19] Current clinical guidelines and funding rules require specific assessments before treatment can be started[20] necessitating multiple clinic visits. Data from the Australian Needle and Syringe Programme Survey estimates that among the 93 000 PWID in Australia, the majority (89%) had a lifetime history of HCV antibody testing.[21] However, individuals are gradually lost throughout the cascade of care. Of Australian Needle and Syringe Programme Survey participants reporting testing positive for an HCV antibody test, only 32% reported accessing HCV treatment up to 2020 (prior to the COVID-19 pandemic).[22]

Point-of-care (POC) HCV testing can simplify the diagnosis of HCV in healthcare settings. The emergence of rapid POC testing for both HCV antibodies and RNA has made this feasible. The OraQuick rapid HCV antibody test is self-contained and uses fingerstick capillary blood samples to provide results in around 20 min. The GeneXpert HCV Viral Load test detects HCV RNA using fingerstick capillary blood to provide results in around 60 min. These tests have been shown to be highly sensitive and specific,[23 24] and overcome many of the barriers to treatment associated with the conventional care model.

Studies suggest POC testing is acceptable among PWID and can be feasibly integrated into health services.[19 25–27] However, limited data have assessed the impact of POC testing on treatment uptake. One study in France found a larger proportion of individuals who received a POC test completed HCV screening and were linked to care compared with those with standard serological testing.[28] Provisional data in Australia found POC testing through needle and syringe exchange programmes reduced time to treatment.[29] In Connecticut, patients selecting POC testing were more likely to be linked to HCV treatment than those opting for standard laboratory testing.[30] These studies suggest that POC testing has a positive effect on retention in HCV care.[27 29] Beyond rapid testing, the feasibility of same-day testing and treatment with DAAs has been demonstrated: preliminary studies have shown high levels of linkage to care following same-day testing and treatment.[31] Same-day testing and treatment have also been shown to achieve high cure rates compared with facilitated referral.[32] However, only a limited number of studies have assessed the impact on retention in care of same-day treatment following POC testing for HCV.[31]

From these existing data, there are broadly three rapid testing models of care being contemplated and deployed in real-world settings: first, POC antibody testing alone (corresponding to arm A of QuickStart Trial, more fully defined below); second, combined POC antibody and RNA testing (arm B); and third, combined POC testing and same-day treatment (Arm C). While promising in pilot studies, none of these models have been tested formally in clinical trials. Clinical trial evidence is required to determine the effectiveness of POC testing with or without treatment compared with the current standard of clinical care.

The QuickStart Trial will first determine the effectiveness of rapid POC testing in primary healthcare settings on treatment uptake. We hypothesise that rapid antibody testing alone (arm A), and rapid antibody followed by immediate RNA testing (arm B), will increase treatment uptake compared with standard clinical care (the control arm). Second, QuickStart will determine the effectiveness of POC antibody testing followed by same-day treatment (Arm C) on cure. We hypothesise that testing and same-day treatment will increase cure compared with standard clinical care. Our secondary outcomes will also compare each of the three intervention models to each other for any incremental benefit on treatment uptake and cure.

## METHODS AND ANALYSIS
### Trial design
A clustered randomised cross-over trial with six treatment sequences will evaluate the effects of rapid POC testing on retention of participants with HCV care and achievement of cure. Each site will have a standard of care (SOC) control period randomly occurring before or after one of three intervention arms, allowing for intrasite comparison. Each site will have an SOC control period randomly occurring before or after one of the three intervention arms, allowing for intrasite and intersite comparisons to be combined in the estimation of treatment effects. The cross-over design has advantages in ensuring that any external changes that occur over time can be accounted for, which a simple before and after intervention comparison would not allow. External changes might occur at

the clinic level, such as services and staff becoming more aware or skilled in offering HCV testing, or they may occur at a system level, such as changes in access to testing, care or treatment. There will be a 1-month washout period between arms at each study site. Participant recruitment began in March 2022 and is intended to continue for up to 24 months.

### Patient and public involvement

We engaged the University of New South Wales to convene a community reference panel to provide input and feedback into the design of the research and study procedures. The panel involved 10 members from the study target audience who were involved in two separate consultations in 2021.

### Study setting

The trial will be conducted at primary healthcare services in metropolitan and rural settings in Australia with:
► High caseloads of people who currently or previously injected drugs and are at risk of HCV infection.
► A general practitioner and/or a nurse practitioner experienced in testing and treating HCV.

### Recruitment

Eligible clinics will be identified through consultation with government, community, clinical and research networks in viral hepatitis in Australia. Within clinics, potentially eligible participants will be approached by a clinician, nurse or community health worker. Participant-facing study promotion including posters and leaflets will be used at the discretion of participating clinics.

### Participant eligibility

Study inclusion criteria for participants are as follows:
► Current or former injecting drug use (ie, injected drugs at least once).
► Aged 18 years or over.
► Attending a participating primary healthcare for any reason.
► No previous treatment with DAAs.
► Medicare eligible.
► Able to speak and understand English.
  Exclusion criteria are as follows:
► Women known to be currently pregnant or who are breast feeding.
► Individuals currently engaged in HCV treatment.
► Unable to provide informed consent.
► Tested for HCV in the past 3 months.
► Previous successful treatment with interferon-based therapy for HCV.

### Study arms

This study will implement three intervention arms and a control arm, as shown in figure 1. Final recruitment, retention and outcome assessments will be presented in accordance with the Consolidated Standards of Reporting Trials (CONSORT) statement, including CONSORT diagrams. Arm A will implement a rapid POC antibody

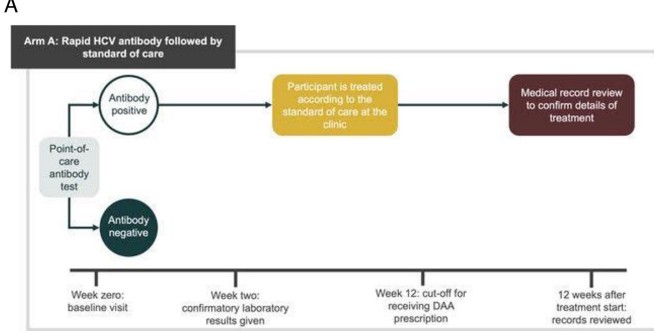

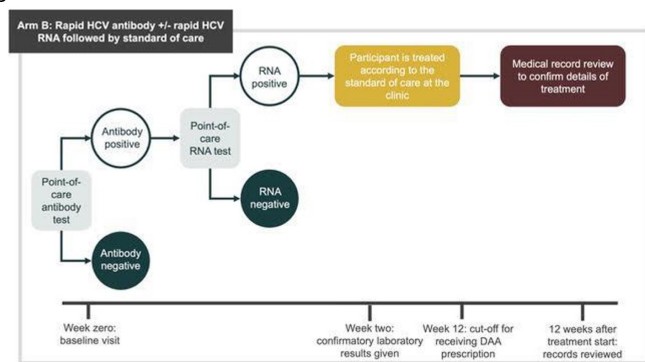

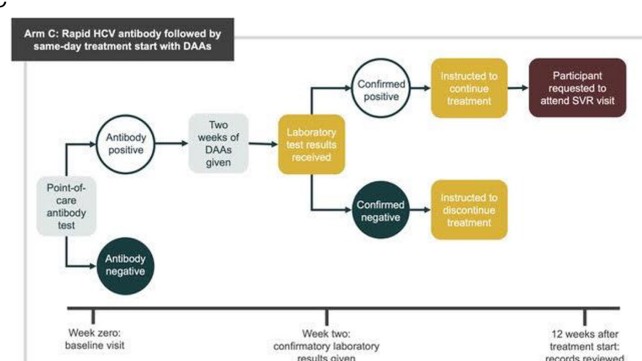

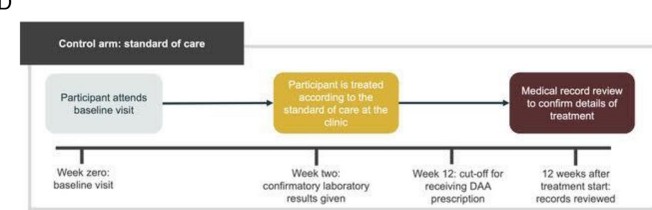

**Figure 1** (A) Sequence of events in arm A: rapid HCV antibody testing followed by standard of care. (B) Sequence of events in arm B: rapid HCV antibody±rapid HCV RNA followed by standard of care. (C) Sequence of events in arm C: rapid HCV antibody followed by same-day treatment with DAAs. (D) Sequence of events in the control arm: standard of care. DAAs, direct-acting antivirals; HCV, hepatitis C virus.

test in addition to SOC confirmatory RNA testing and blood tests at the baseline visit. Arm B will implement a rapid POC antibody and POC RNA test in addition to SOC blood tests at the baseline visit. Arm C will implement a rapid POC antibody test and same-day treatment with DAAs for antibody-positive participants, in addition to standard blood tests. The control arm will implement the SOC in operation at each clinic.

All participants will be administered a clinical and behavioural questionnaire at their baseline visit. All arms are outlined in detail below.

### Arm A: rapid POC antibody+SOC
*Baseline visit*
► Participants will receive an OraQuick rapid HCV antibody test using a fingerstick specimen, according to the manufacturer's instructions, with the result documented at 20 min. The result will be provided to the participant as a preliminary result with laboratory testing on a venous sample required to confirm the POC test result. If the POC result is positive, blood samples will be collected for any confirmatory laboratory antibody and RNA testing and SOC pretreatment screening. If the POC result is negative, confirmatory testing will be offered but the participant is given the option to decline.

*Week 0–2 (confirmatory lab results)*
► If the confirmatory laboratory HCV antibody result is negative, the nurse will communicate this to the participant. The participant will complete the study here. Standard harm minimisation messaging and window periods for detection of HCV will be discussed with the participant.
► If the confirmatory laboratory HCV antibody result is positive and the RNA result is negative, the nurse will contact the participant by phone to inform them of their result—prior exposure to HCV, no current infection. The participant will complete the study here.
► If the confirmatory laboratory HCV antibody result is positive and the RNA result is positive, the nurse will contact the participant by phone to inform them of their result of current HCV infection and arrange a follow-up appointment to discuss treatment as per the clinic usual SOC.

*Weeks 2–12 (HCV RNA positive)*
► HCV RNA positive participants will be followed up as per usual clinic SOC.

*Week 12 (HCV RNA positive)*
► The study nurse will review the medical records of consented participants to determine whether participants with HCV were provided with a DAA prescription within 12 weeks of enrolment in the study.
► All participants who have been provided with a DAA prescription will be contacted by the study nurse to ask whether they started treatment and if so, when.

*12 weeks following treatment completion (sustained virological response or cure assessment):*
► The medical records of all participants confirmed through medical record review to have started DAA treatment will be reviewed to determine if they have had a sustained virological response (SVR) blood test anytime from 4 to 12 weeks following their treatment completion date.[33] Reinfection will be differentiated from treatment failure due to relapse by assessing (1) genotype or subtype and (2) viral sequencing at a reference laboratory, whenever possible.
► Participants who have not had an SVR visit will be contacted by the study nurse approximately 12 weeks following the scheduled treatment completion date (determined by the date they received their script and duration of treatment) to request that they attend an SVR visit.

### Arm B: rapid POC antibody+rapid POC RNA+SOC
*Baseline visit*
► Participants will receive an OraQuick rapid HCV antibody test using a fingerstick specimen, according to the manufacturer's instructions, with the result documented at 20 min. The result from the 20 min time point will be provided to the participant as a preliminary result, with laboratory testing on a venous sample required to confirm. Blood samples will be collected for any confirmatory laboratory testing and SOC pretreatment screening.
► Participants with a negative OraQuick result will be provided with this preliminary result and informed that they will be contacted within the next 2 weeks with any laboratory test results.
► Participants with a positive OraQuick result will be offered POC RNA testing using the Cepheid GeneXpert HCV RNA fingerstick test. The study nurse will conduct POC RNA testing using a fingerstick blood sample according to the manufacturer's instructions, with results in ~60 min. The study nurse will determine with the participant how they would like to receive these results—via remaining on site or by phone call. If the HCV POC RNA is negative, the study nurse will inform the participant of their result, either in person or by telephone, and they will be contacted within the next 2 weeks with confirmatory laboratory test results.
► If the HCV POC RNA is positive, the study nurse will inform the participant of their result either in person or by telephone and they will be contacted within the next 2 weeks with confirmatory laboratory test results.

*Weeks 0–2 (confirmatory lab results)*
► If the confirmatory laboratory RNA results are negative, the nurse will inform the participant and the participant will complete the study. Standard harm minimisation messaging and window periods for detection of HCV will be discussed with the participant.
► If the laboratory HCV RNA is positive, the study nurse will contact the participant to provide their results of current HCV infection and discuss follow-up and treatment according to the practice of the clinic.

*Weeks 2–12 (HCV RNA positive)*
► Participants with HCV will be followed up as per usual clinic SOC.

### Week 12 (HCV RNA positive)

► The study nurse will review the medical records of consented participants to determine whether participants with HCV were provided with a DAA prescription within 12 weeks of enrolment in the study.

► All participants who have been provided with a DAA prescription will be contacted by the study nurse to ask whether they started treatment and if so, when.

### 12 weeks following treatment completion (SVR)

SVR will be assessed as previously described for arm A.

### Arm C: same-visit rapid POC antibody+treatment commencement
### Baseline visit

► Participants will receive an OraQuick rapid HCV antibody test using a fingerstick specimen, according to the manufacturer's instructions, with the result documented at 20 min. The result from the 20 min time point will be provided to the participant as a preliminary result, with laboratory testing on a venous sample required to confirm. Blood samples will be collected for any confirmatory laboratory testing and SOC pretreatment screening.

► Participants with a negative OraQuick result will be advised that the POC test result will be confirmed by standard pathology results and they will be contacted in 1–2 weeks with these results.

► Participants with a positive OraQuick result will be provided with 2 weeks of sofosbuvir/velpatasvir (Epclusa) and will be instructed to begin treatment. In this sense, arm C uses presumptive treatment, where 14 days of treatment will be dispensed prior to laboratory confirmed diagnosis.[34] Blood samples will be sent for confirmatory HCV antibody and RNA testing. Participants of childbearing potential will be screened for possible pregnancy and if indicated, a urinary pregnancy test will be performed before dispensing Epclusa. If the pregnancy test is positive, they will not be provided with a study drug and will be referred to SOC. Medical records will be reviewed for potential drug-drug interactions with Epclusa, and participants will be asked about medications they are currently taking. Participants taking medications with red or orange drug–drug interactions will not be provided with a study drug and will be referred to SOC.

### Weeks 1–2 (confirmatory results)

► Participants will be contacted with the results of their confirmatory laboratory blood tests. Participants with a positive confirmatory antibody and negative RNA test result will be contacted with this result of prior exposure, no current infection and instructed to discontinue treatment and dispose of any unused medications. They will complete the study here. Standard harm minimisation messaging and window periods for detection of HCV will be discussed with the participant.

► Participants with a positive confirmatory antibody and positive confirmatory RNA result will be contacted to arrange ongoing treatment through the study, a further 4 weeks of sofosbuvir/velpatasvir will be dispensed. Further assessment may be required based on the laboratory blood test results:

 – If the participant received an Aspartate Animotransferaese (AST):Platelet Ratio Index score of <1.0, they will be instructed to continue treatment as they were originally instructed.

 – If the participant received an AST:Platelet Ratio Index score of ≥1.0, they will be instructed to continue taking the DAAs and will be referred to a specialised.

 – Participants found to have renal impairment (estimated glomerular filtration rate (eGFR) <30 mL/min) or coinfected with chronic hepatitis B will be instructed to cease treatment and will be referred to a specialist.

 – Participants found to have HIV infection will be instructed to continue the DAAs and be referred to a specialist.

Following specialist assessment, the participant may be required to change medications or alter their treatment course, according to the specialist's recommendations.

### Week 6 (HCV RNA positive)

► Participants on treatment will be asked to attend a follow-up appointment where they will be provided with a further 4 weeks of sofosbuvir/velpatasvir.

### Week 10 (HCV RNA positive)

► Participants who have received 10 weeks of treatment will be contacted to arrange a follow-up appointment to receive the final 2 weeks of treatment.

### 12 weeks following treatment completion (SVR)

SVR will be assessed as previously described for arm A.

### Control arm: SOC testing and treatment

All participants will be tested, treated and followed up according to the usual practice of the clinic for delivery of test results and ongoing HCV treatment. The participants' medical record will be reviewed at the study close to assess testing and treatment uptake.

### Study medication

For arms A, B and the control, medications will be funded by the PBS and dispensed utilising standard pharmacy practices. The type of DAAs used to treat participants with HCV in these arms will be at the discretion of the clinical service where the participant is enrolled. For arm C, study medications will be stored and dispensed by participating study sites. The medication used in arm C of this study is a 12 week course of oral combination therapy sofosbuvir/velpatasvir (Epclusa).

Epclusa tablets contain 400 mg of sofosbuvir and 100 mg of velpatasvir, to be taken orally once daily. Epclusa is TGA approved with ARTG ID 266823.[35]

## Assignment of interventions

Intervention arms will be assigned to sites using random number generation in a centralised system. Sites will be randomised in batches of six to ensure a balance of the number of sites in each arm over time. Sites will be informed of their allocation to each of the two consecutive arms separately. Due to the nature of the study design, it is not possible to blind clinics or participants during all phases of the intervention. All sites randomised to arms A, B or C in the first period will know that they will be allocated to the control arm after the cross-over period, due to the nature of the study design. Sites allocated to the control arm in the first period will be blinded as to which arm they will be randomised to, following the cross-over washout period (see figure 1).

## Study outcomes

The two primary outcomes of this trial include:

1. The proportion of participants commencing HCV treatment in each arm. Commencement of HCV treatment is defined as receipt of a script for treatment in arms A, B and control and as commencement of treatment with study drug in arm C. The proportion commencing treatment in each intervention arm will be compared with the control arm.
2. The proportion of participants with HCV who start treatment achieving cure. Cure will be assessed by review of medical records for a clinic visit to assess SVR; if no visit has taken place, the study nurse will contact the participant to arrange one in all arms except the control.

Commencing treatment is an important and easily measurable outcome in clinical practice. The rationale for two primary outcomes is that arm C itself involves providing treatment to everyone found to have HCV as part of its intervention.

Secondary outcomes of this study include:

1. Time to treatment commencement is defined as number of days from first test to first dose of treatment.
2. Case finding is defined as the proportion of people diagnosed with HCV among all those recruited.
3. Cost-effectiveness is defined as the incremental cost to find and commence treatment among one person living with HCV from a health system perspective.

Participant acceptability is based on qualitative evaluation subject to further substudy design and ethical approval.

## Sample size

Comparisons of primary interest are comparing outcome 1 (treatment commencement) in control and arm A; outcome 1 in control and arm B; and outcome 2 (HCV cure) in control and arm C. We consider sample size calculations with a minimum of 80% power and a significance level of 0.05/3=0.0167. Comparisons of control versus A are expected to result in the smallest required sample sizes so sample size calculations focus on this comparison. We use the Shiny CRT app[36] for sample size calculations,

assuming that the t-distribution is used instead of the normal approximation (accounting for small numbers of clusters).[37]

We suppose that five sites are assigned to each of the six treatment sequences. We found that with 24 participants recruited in each cluster in each period, power will be greater than 80% to detect a change in proportions of participants starting HCV treatment of 30%–50% where 10 clusters are allocated to arm A. Due to the greater portion of participants expected to start treatment in arm B (70%) and arm C (90%), power >80% is achieved with eight clusters assigned to the sequences that involve arms B or C (ie, with four clusters randomised to each of the arm B then control; control then arm B; Arm C then control; and control then arm C sequences). The proportion of participants expected to achieve cure in each arm are 10% among controls, 30% in arm A, 50% in arm B and 70% in arm C. Allowing for 20% loss to follow-up of sites and of participants, 30 participants per site per period would need to be recruited. Given these results, we will instruct sites to aim for 40 participants per site per period to counteract instances where some sites are unable to achieve 30 participants per site per period; some sites achieving 40 participants will balance sites which achieve <30. This would imply a maximum total of 2400 participants would need to be recruited.

## Statistical analysis

Results from the trial will be reported according to the CONSORT extension statement for cluster randomised trials,[38] or a CONSORT extension statement for cluster randomised crossover trials if one is available prior to end of data collection. Clinic and patient demographics will be described by site and study period. Analyses will be conducted on an intention-to-treat basis, with patients analysed according to the randomised treatment of their cluster at the time of their inclusion in the study. All available binary outcomes will be analysed at the patient level using mixed-effects logistic regression models with random intercepts for clinic and clinic period and fixed effects for interventions.[39] ORs and 95% CIs will be reported; in addition, we will calculate and report risk differences and 95% CIs.

There are three main comparisons proposed, namely, treatment uptake outcome in arm A versus control (comparison 1) and arm B versus control (comparison 2) and HCV cure in Arm C versus control (comparison 3). Incremental benefits of arm B to arm A, and arm C to arm B, and arm C to arm A on treatment uptake and HCV cure will be performed in subgroup analyses. Data from controls in each arm will be pooled and a single model fitted for each outcome.

Randomisation will occur in batches of six, performed by the coordinating centre and concealed from investigators and sites until after site governance approcal. Since randomisation is occurring within batches of six services, fixed effects for batch will be included in these models. The Kenward-Roger correction will be applied

to adjust for the small number of clinics.[40] Estimates of intracluster correlations (within-period and between-period) will be reported. Time-to-event outcomes such as time to complete diagnosis will be analysed using Cox proportional hazards models with shared-frailty terms for site and fixed terms for treatment group and randomisation batch. There may be imbalances between sites with respect to various clinic and patient-level characteristics, and secondary analyses will adjust for these.

Sensitivity analyses to investigate the impact of different assumptions about missing data on study conclusions will be performed. Initially, two scenarios will be considered: 'best-worst' and 'worst-best'. In the 'best-worst' scenario, all patients with missing data in clinics during periods of administration of arms A, B or C will be assumed to have commenced HCV treatment, and all patients in control periods will be assumed not to have commenced HCV treatment. In the 'worst-best' scenario, all patients with missing data in clinics during periods of administration of arms A, B or C will be assumed to not have commenced HCV treatment, and all patients in control periods will be assumed to have commenced HCV treatment. Similar scenarios for the SVR outcome will be considered. Should these analyses lead to different study conclusions, further analyses including terms predictive of missingness will be conducted or using multiply imputed datasets will be considered.

## ETHICS AND DISSEMINATION
### Ethical review
This study has been approved by the Alfred Ethics Committee (Number: HREC/64731/Alfred-2020-217547). Informed consent will be obtained from all participants prior to enrolment. Potential participants will be presented with detailed information about the study procedures, objectives, risks and benefits. Participants will be able to refuse or terminate participation at any time. Participants will be reimbursed US$40 at their baseline visit and US$80 for their SVR visit. Reimbursement will be by Coles/Myer voucher or by direct funds bank transfer (EFT/PayID).

### Data monitoring
The trial will be monitored by an independent data safety and monitoring board with an independent statistician. There will be predefined periodic analysis to assess safety endpoints (drug resistance among treatment commencing participants, treatment failures, serious adverse events as detailed below) in accordance with a statistical analysis plan under development before conclusion of recruitment. Access to identified study data will be limited to the data manager and study coordinator. Deidentified data will be limited to study investigators.

### Adverse events
This study uses a drug with side effects that are already known. It will, therefore, report adverse events that, in the opinion of the investigator, are not deemed to be expected or are increased in severity from prior observations. Side effects not already known of the drug in a severity that the treating investigator deems as unexpected/serious should be reported to Burnet Institute as sponsor.

For this study, we will only be collecting adverse events that are inconsistent with prior observations reported in study drug product information. Adverse events considered to be related to study drug administration will be captured up until 24 weeks after the last recorded study dose.

Serious adverse events will be reported to the Burnet as sponsor within 24 hours of the site learning of them and include any event that results in death; threat to life; significant disability or incapacity; congenital abnormality or birth defect; hospitalisation or is medically significant and requires intervention to prevent any of the former outcomes.

Burnet Institute is responsible for reporting safety events to the reviewing HREC in accordance with Safety Monitoring and Reporting in Clinical Trials Involving Therapeutic Goods (NHMRC, 2016).

In line with the Alfred Hospital Ethics Committee Safety Monitoring and Reporting Requirements (October 2017), only events that meet the criteria of a significant safety issue and/or urgent safety measure will need to be reported to the Alfred HREC.

### Dissemination
Publication of data derived from the study will be supervised by the protocol steering committee. The results of the project will be published and presented at scientific meetings. All published quantitative data will be non-identifiable grouped data. Proposals for additional projects or collaborations, likely to result in separate publications, will be reviewed by the Steering Committee for final determination. A plain English summary of study outcomes, as well as abstracts from publications, will be available on Burnet Institute website. The plain English summary of study outcomes will also be provided to all participating sites. Results may also form part of PhD or honours student theses. Authorship for publications arising from this study will adhere to the International Committee of Medical Journal Editors guidelines.[41]

**Author affiliations**
[1]Infectious Diseases, Monash University, Melbourne, Victoria, Australia
[2]Burnet Institute, Melbourne, Victoria, Australia
[3]Population Health, Monash University, Melbourne, Victoria, Australia
[4]Gastroenterology, St Vincent's Hospital, Melbourne, Victoria, Australia
[5]Department of Medicine at St Vincent's Hospital, The University of Melbourne, Melbourne, Victoria, Australia
[6]Wesfarmers Centre of Vaccines and Infectious Diseases, Telethon Kids Institute, Nedlands, Western Australia, Australia
[7]Faculty of Medicine and Health, The University of Sydney, Sydney, New South Wales, Australia
[8]HepatitisWA, Perth, Western Australia, Australia
[9]Public Health, La Trobe University, Bundoora, Victoria, Australia

**Contributors** Conception: JSD, MH and AP. Design and development: JSD, KH, IE, AW, JK, KA, MB, AT, TLS, TS, PD, PH, AF, MH and AP. Conduct and acquisition of data: JSD, KH, IE, CD, AW, KA, SVB, KC, BC, RG, MB, JR, NA, PH and AP. Analysis and interpretation: JSD, KH, JK, AT, MAS, TLS, NS, TS, DA, RS-D, JH, AF, MH and AP. Drafting, critical review and approval of manuscript: JSD, KH, IE, CD, AW, JK, KA, SVB, KC, BC, RG, MB, AT, MAS, TLS, NS, TS, DA, JR, JH, NA, PD, PH, RS-D, AF, MH and AP. Guarantor: JSD, AP (co-principal investigators). JSD and KH contributed equally to this manuscript.

**Funding** QuickStart is support by the National Health and Medical Research Council (GNT1188026) and an investigator-initiated grant from Gilead Sciences (no grant number).

**Competing interests** JSD, MH and AP report investigator-iniated funding to their institution from Gilead Sciences, AbbVie and Merck. JSD reports honoraria to his institution from Gilead Sciences and AbbVie.

**Patient and public involvement** Patients and/or the public were involved in the design, or conduct, or reporting, or dissemination plans of this research. Refer to the Methods section for further details.

**Patient consent for publication** Not applicable.

**Provenance and peer review** Not commissioned; externally peer reviewed.

**ORCID iDs**
Joseph S Doyle http://orcid.org/0000-0001-7198-0833
Jessica Kasza http://orcid.org/0000-0002-8940-0136
Alisa E Pedrana http://orcid.org/0000-0002-1998-5722

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
