## [Reviewer comments · BMJ Open]

ARTICLE DETAILS

TITLE (PROVISIONAL)	Same-visit hepatitis C testing and treatment to accelerate cure among people who inject drugs (The QuickStart Study): a cluster randomised crossover trial protocol
AUTHORS	Doyle, Joseph; Heath, Katherine; Elsum, Imogen; Douglass, Caitlin; Wade, Amanda; Kasza, Jessica; Allardice, Kate; Von Bibra, Sally; Chan, Kico; Camesella, Beatriz; Guzman, Rodney; Bryant, Mellissa; Thompson, Alexander; Stoové, Mark; Snelling, Thomas; Scott, Nick; Spelman, Tim; Anderson, David; Richmond, Jacqui; Howell, Jess; Andric, Nada; Dietze, Paul; Higgs, Peter; Sacks-Davis, Rachel; Forbes, Andrew; Hellard, Margaret; Pedrana, Alisa

VERSION 1 – REVIEW

REVIEWER	Cristina Murray-Krezan University of Pittsburgh Department of Medicine
REVIEW RETURNED	08-Jun-2023

GENERAL COMMENTS	This manuscript presents the protocol for a cluster-randomized crossover trial with 4 arms to assess point-of-care testing and treatment for HCV. The trial has already started. The following are significant points that should be addressed in order to improve readers' understanding of the rationale for the study design and proposed analysis of the study data: ●The rationale for the design of the study is not completely clear to me. It seems like offering POC testing and POC treatment is an obvious comparator to, perhaps, all of the other arms, but it is not clear the benefit of having the other active arms without explicit primary outcome-level testing between the arms, potentially for non-inferiority or what ever the hypotheses may be. The hypotheses are not explicitly stated.●The primary endpoints help to rationalize the study design, but should be discussed earlier in the protocol/manuscript. However, primary outcome 1 appears to be testing each of the active arms to the control, but no comparisons to each other. Presumably this will be a(n) (undefined) secondary outcome? Are the authors pooling across all of the controls when comparing a given tx arm to control or just that tx arm's control? Please further describe the rationale for the cross-over design.●While a Bonferroni adjustment to the type I error was made for the sample size estimate of the first primary outcome, there was no accounting for multiplicity for two primary outcomes (and really four primary endpoints). Additionally, no hypothesized/estimated effect size is reported for the secondary outcome (cure in Tx C vs. Control)
--

	given the sample size determined from primary outcome 1. This should be reported. What are the hypothesized/detectable effect sizes for Tx B & C vs. control for DAA commencement? These should also be reported.  ●The primary endpoints should be defined along with time points and windows for measurement. For participants who need to be contacted by the study nurse for a 12-week SVR visit, what is the window for their visit? How will the authors minimize the risk of actually measuring reinfection rather than not reaching SVR? ●In the analysis section, there is a description of fitting mixed effects logistic regression models with random effects for clinic and clinic period. Shouldn't participant also be included as a random effect? Along with this, reference 40 (the Kenward-Roger correction to adjust for small number of clinics) is a correction for fixed effects but clinic is a random effect, so please justify why this correction will be applied to clinic. Also, is 30 clinics considered a small number of clinics? I am surprised by this. Lastly, odds ratios should be calculated/reported when the probability of the event is small, otherwise the effect is overestimated. The probability of cure given treatment is not at all small (>90%, as stated by the authors in the Introduction) and since the study team is assessing cure given the participant started treatment, one would expect the cure rates to be similar and high in the Tx A and control arms. The authors reported the change in the proportion of people hypothesized to start DAA, but did not mention what the comparator baseline DAA commencement rate is. It is mentioned in the Introduction that 32% of HCV-positive participants in a survey reported accessing treatment in 2020. This was during the peak of the pandemic, so if this is the rate being used for the control group, I wonder if it would be better to choose a rate from 2019 for less bias (unless there are policy reasons to report from 2020). Nevertheless, assuming 32% commencement rate in the control, that is still not really a rare event and ORs may overestimate effects. Consider exploring different measures of rates to report. ●When describing the study design, please remove the redundant "cross-over" in the Abstract and Trial Design sections (should just be "A cluster randomised crossover trial"). ●The reviewer appreciates the detailed study flow diagrams for each arm, but strongly recommends additionally including a standard CONSORT diagram. ●Please fix formatting issues including the caption overlapping the diagram in Figure 1 and the numbering of the sections of the manuscript. Also proofread for typos and grammatical errors--I saw several. ●I note that the authors stated in the Dissemination section that "The results of the project may be published..." and I strongly encourage "may" be changed to "will". All results from clinical trials should be published to minimize redundancy and waste in research.
--	--

REVIEWER	Biagio Pinchera University of Naples Federico II
REVIEW RETURNED	17-Jun-2023

GENERAL COMMENTS	I believe that the study by Doyle J et al could be a milestone in the
---

	management of HCV infection in PWID patients. As the authors allow to evaluate potential strategies to improve and implement the activity of screening, linkage to care and treatment of HCV infection. This study appears to be very interesting and innovative with potential and possible new diagnostic-therapeutic approaches in the management of HCV infection in PWID patients. For this reason I believe that this article can be published. I would ask the authors to cite the following articles:  - Di Minno MND, Ambrosino P, Buonomo AR et al. Direct-acting antivirals improve endothelial function in patients with chronic hepatitis: a prospective cohort study. Intern Emerg Med. 2020 Mar;15(2):263-271. doi: 10.1007/s11739-019-02163-8. - Coppola N, Portunato F et al. Interferon-free regimens improve kidney function in patients with chronic hepatitis C infection. J Nephrol. 2019 Oct;32(5):763-773. doi: 10.1007/s40620-019-00608-z. - Gentile I, Fusco F et al. Prevalence and risk factors of erectile dysfunction in patients with hepatitis B virus or hepatitis C virus or chronic liver disease: results from a prospective study. Sex health. 2018 Nov;15(5):408-412. doi:10.1071/SH17168. Thank you!
--	--

VERSION 1 – AUTHOR RESPONSE

Reviewer #1’s comments

Dr. Cristina Murray-Krezan, University of Pittsburgh Department of Medicine

1. *The rationale for the design of the study is not completely clear to me. It seems like offering POC testing and POC treatment is an obvious comparator to, perhaps, all of the other arms, but it is not clear the benefit of having the other active arms without explicit primary outcome-level testing between the arms, potentially for non-inferiority or what ever the hypotheses may be. The hypotheses are not explicitly stated.*

We have amended the final introductory paragraph to clarify the rationale and hypotheses (at page six, line 126):

“From these existing data, there are broadly three rapid testing models of care being contemplated and deployed in real-world settings: firstly, POC antibody testing alone (corresponding to Arm A of QuickStart, more fully defined below); secondly, combined POC antibody and RNA testing (Arm B); and thirdly, combined POC testing and same-day treatment (Arm C). While promising in pilot studies, none of these models have been tested formally in clinical trials. Clinical trial evidence is required to determine effectiveness of POC testing with or without treatment compared with the current standard of clinical care.

The QuickStart Trial will firstly determine the effectiveness of rapid POC testing in primary health care settings upon treatment uptake. We hypothesise that rapid antibody testing alone (Arm A), and rapid antibody followed by immediate RNA testing (Arm B), will increase treatment uptake compared with standard clinical care (the control arm). Secondly, QuickStart will determine the effectiveness of POC antibody testing followed by same-day treatment (Arm C) upon cure. We hypothesis that testing and same-day treatment with increase cure compared with standard clinical care. Our secondary outcomes will also compare each of the three intervention models to each other for any incremental benefit on treatment uptake and cure.”

- 2. The primary endpoints help to rationalize the study design, but should be discussed earlier in the protocol/manuscript. However, primary outcome 1 appears to be testing each of the active arms to the control, but no comparisons to each other. Presumably this will be a(n) (undefined) secondary outcome?*

Thank you, we have added rationale in the introduction (see point one, above) before the description of the endpoints in the methods section.

We wish to clarify that there are **two primary outcomes** (treatment uptake, and HCV cure) as already specified in “Methods: Study outcomes” at page 18, line 392..

While commencing treatment is easier to measure and the important outcome in clinical practice, the rationale for two outcomes is that Arm C itself involves providing treatment to everyone found to have HCV, so it cannot be compared to primary outcome one (treatment uptake). This has been explicitly stated for clarity at page 19, line 402.

However, there are **three main comparisons proposed**, namely, treatment uptake outcome in Arm A versus control (comparison one) and Arm B versus control (comparison two), and HCV cure in Arm C versus control (comparison three). A specific comment for clarity is added in the “Methods: statistical analysis” section at page 21, line 449 in addition to the existing comment in the “Methods: sample size” section at page 19, line 416.

We have added a comment about specific sub-group analyses to make comparisons to between each of the arms to each other to measure any incremental benefit (at page 21, line 451).

Finally, for transparency we have specified other planned **secondary outcomes** (at page 19, line 405) including time to treatment commencement, case finding, cost-effectiveness, and participant acceptability.

- 3. Are the authors pooling across all of the controls when comparing a given tx arm to control or just that tx arm's control?*

Our design and analysis plan will permit controls in each arm to be pooled and single model fitted for each comparison.

Text has been added to “Methods: statistical analysis” (at page 21, line 453):

“Data from controls in each arm will be pooled and a single model fitted for each outcome.”

4. *Please further describe the rationale for the cross-over design.*

We added text to explain the crossover design and rationale in "Methods: trial design" section (at page seven, line 155):

“Each site will have a standard of care control period randomly occurring before or after one the three intervention arms, allowing for intra-site and inter-site comparisons to be combined in the estimation of treatment effects. The cross-over design has advantages in ensuring that any external changes that occur over time can be accounted for, which a simple before and after intervention comparison would not allow. External changes might occur at the clinic level, such as services and staff becoming more aware or skilled in offering hepatitis C testing, or they may occur at a system level, such as changes in access to testing, care or treatment.”

5. *While a Bonferroni adjustment to the type I error was made for the sample size estimate of the first primary outcome, there was no accounting for multiplicity for two primary outcomes (and really four primary endpoints). Additionally, no hypothesized/estimated effect size is reported for the secondary outcome (cure in Tx C vs. Control) given the sample size determined from primary outcome 1. This should be reported. What are the hypothesized/detectable effect sizes for Tx B & C vs. control for DAA commencement? These should also be reported.*

We have calculated our sample size based on the smallest hypothesised difference, that is between intervention arm A and control accounting for the three primary comparisons by assuming a two-sided significance level of $0.05/3 = 0.0167$ (see “Methods: sample size” at page 19, line 416).

As clarified in response to the reviewer’s second point above, there are only three main comparisons (ie Arm A and Arm B comparisons with control for outcome/endpoint one, and Arm C comparison with control for outcome/endpoint two).

We have added estimates of the proportions of patients experiencing each outcome under usual care for all arms, and treatment uptake assumptions for Arm B and C as suggested in “Methods: sample size” (at page 20, line 426):

“Due to greater portion of participants expected to start treatment in Arm B (70%) and Arm C (90%), power >80% is achieved with eight clusters assigned to the sequences that involve Arms B or C (i.e. with 4 clusters randomised to each of the Arm B then Control; Control then Arm B; Arm C then Control; and Control then Arm C sequences). The proportion of participants expected to achieve cure in each arm are 10% among controls, 30% in Arm A, 50% in Arm B, and 70% in Arm C.”

6. *The primary endpoints should be defined along with time points and windows for measurement. For participants who need to be contacted by the study nurse for a 12-week SVR visit, what is the window for their visit? How will the authors minimize the risk of actually measuring reinfection rather than not reaching SVR?*

We have defined SVR12 measurement periods on page 11 (at line 242):

“12 weeks following treatment completion (Sustained Virological Response (SVR) or cure assessment): The medical records of all participants confirmed through medical record review to have started DAA treatment will be reviewed to determine if they have had a SVR blood test anytime from 4 -12 weeks following their treatment completion date.”

Further, we have now added a clarification at page 11, line 247, that:

“Reinfection will be differentiated from treatment failure due to relapse by assessing (1) genotype or subtype, and (2) viral sequencing at a reference laboratory, whenever possible.”

7. *In the analysis section, there is a description of fitting mixed effects logistic regression models with random effects for clinic and clinic period. Shouldn't participant also be included as a random effect? Along with this, reference 40 (the Kenward-Roger correction to adjust for small number of clinics) is a correction for fixed effects but clinic is a random effect, so please justify why this correction will be applied to clinic. Also, is 30 clinics considered a small number of clinics? I am surprised by this.*

Since each participant provides a single measurement only for each outcome (i.e. separate individuals are recruited in each cluster in each period of the study), there is no need to include participant-level random effects.

Thirty clinics is indeed considered to be a small number of clinics (or clusters).

Our interest here is in making inference about the treatment effect, which is included as a fixed effect in the mixed effects model for each outcome. When the number of clusters is large enough for asymptotic results to apply (where large is generally considered to be 40 or 50 clusters, but there is debate about this in the literature), the sampling distribution for these fixed effects can be well approximated. However, when the number of clusters is small, asymptotic results do not apply, and thus the F distribution with a certain denominator degrees of freedom provides a better approximation. The Kenward-Roger correction is one way of computing this denominator degrees of freedom, and is frequently used in the context of cluster randomised trials, see for example:

- Leyrat C et al. Cluster randomized trials with a small number of clusters: which analyses should be used? *International Journal of Epidemiology*. 2018;47:321-331.
- Thompson JA et al. Cluster randomised trials with a binary outcome and a small number of clusters: comparison of individual and cluster level analysis method. *BMC Medical Research Methodology*. 2022;22:222.

Note that these references also refer to the number of clusters less than or equal to 30 as being a small number of clusters.

8. *Odds ratios should be calculated/reported when the probability of the event is small, otherwise the effect is overestimated. The probability of cure given treatment is not at all small (>90%, as stated by the authors in the Introduction) and since the study team is assessing cure given the participant started treatment, one would expect the cure rates to be similar and high in the Tx A and control arms.*

We agree with the reviewer that when odds ratios are reported, they may often be misinterpreted as risk ratios, and that odds ratios only provide an approximation to risk ratios when the probability of the event is small. In addition to odds ratios, we will also calculate and report risk differences and 95% confidence intervals (with further details to be outlined in our full statistical analysis plan).

We have added in the “Methods: statistical analysis” that risk differences will also be calculated and reported (see page 21, line 447).

9. *The authors reported the change in the proportion of people hypothesized to start DAA, but did not mention what the comparator baseline DAA commencement rate is. It is mentioned in the Introduction that 32% of HCV-positive participants in a survey reported accessing treatment in 2020. This was during the peak of the pandemic, so if this is the rate being used for the control group, I wonder if it would be better to choose a rate from 2019 for less bias (unless there are policy reasons to report from 2020). Nevertheless, assuming 32% commencement rate in the control, that is still not really a rare event and ORs may overestimate effects. Consider exploring different measures of rates to report.*

Baseline treatment uptake of 30% is used in the sample size and is reported in the “Methods: sample size” section.

The 32% proportion quoted has been taken from the 2016-2020 Australian Needle Syringe survey dataset published in 2021. We have clarified that the proportion reported was prior to the pandemic (at page 5, line 102):

“Of ANSPS participants reporting testing positive to an HCV antibody test, only 32% reported accessing HCV treatment annually up to 2020 prior to the COVID19 pandemic.”

As noted above in response to point eight, we will also calculate and report risk differences and associated confidence intervals.

10. *When describing the study design, please remove the redundant "cross-over" in the Abstract and Trial Design sections.*

Thank you, we have made this amendment as suggested.

11. *The reviewer appreciates the detailed study flow diagrams for each arm, but strongly recommends additionally including a standard CONSORT diagram.*

We have added references to CONSORT reporting and the CONSORT checklist as recommended by the Editor. We will follow the CONSORT reporting guidelines of the study outcomes, including a standard CONSORT diagram on the recruitment, retention, and outcomes for each of the four arms in the final results. We have added this text to confirm compliance with CONSORT reporting (at page 9, line 202).

In the absence of results, in this protocol we have elected to leave the pictorial representations of each arm since they provide the reader with a clear understanding of the interventions.

In our judgment, a CONSORT diagram at this stage would not be informative without data on recruitment, retention and achievement of outcomes.

12. *Please fix formatting issues including the caption overlapping the diagram in Figure 1 and the numbering of the sections of the manuscript. Also proofread for typos and grammatical errors.*

Thank you, we have fixed the figure layout and fully edited the document again.

13. *I note that the authors stated in the Dissemination section that "The results of the project may be published..." and I strongly encourage "may" be changed to "will". All results from clinical trials should be published to minimize redundancy and waste in research.*

Thank you, we agree, and have made this amendment.

Reviewer #2's comments

Dr. Biagio Pinchera, University of Naples Federico II

1. *I believe that the study by Doyle J et al could be a milestone in the management of HCV infection in PWID patients. As the authors allow to evaluate potential strategies to improve and implement the activity of screening, linkage to care and treatment of HCV infection. This study appears to be very interesting and innovative with potential and possible new diagnostic-therapeutic approaches in the management of HCV infection in PWID patients. For this reason I believe that this article can be published.*

Thank you for your review and supportive comments. No further specific changes have been made.

VERSION 2 – REVIEW

REVIEWER	Cristina Murray-Krezan University of Pittsburgh Department of Medicine
REVIEW RETURNED	26-Jan-2024
GENERAL COMMENTS	I appreciate the authors' responses and for clarifying the misunderstanding that I had regarding the modeling. I agree with all responses made. There is a typo on line 140 ("with" should be "will") and another on line 150 ("one the" should be "one of the").